# Enhancing Object Detection in Smart Video Surveillance: A Survey of Occlusion-Handling Approaches

**Zainab Ouardirhi** [1,2,*]**, Sidi Ahmed Mahmoudi** [1] **and Mostapha Zbakh** [2]

1 Computer and Management Engineering Department, UMONS Faculty of Engineering, University of Mons, 7000 Mons, Belgium; sidi.mahmoudi@umons.ac.be

2 Communication Networks Department, Ecole Nationale Supérieure d'Informatique and Systems Analysis, Mohammed V University in Rabat, Rabat 10000, Morocco; mostapha.zbakh@ensias.um5.ac.ma

* Correspondence: zainab.ouardirhi@umons.ac.be; Tel.: +32-465582420

**Abstract:** Smart video surveillance systems (SVSs) have garnered significant attention for their autonomous monitoring capabilities, encompassing automated detection, tracking, analysis, and decision making within complex environments, with minimal human intervention. In this context, object detection is a fundamental task in SVS. However, many current approaches often overlook occlusion by nearby objects, posing challenges to real-world SVS applications. To address this crucial issue, this paper presents a comprehensive comparative analysis of occlusion-handling techniques tailored for object detection. The review outlines the pretext tasks common to both domains and explores various architectural solutions to combat occlusion. Unlike prior studies that primarily focus on a single dataset, our analysis spans multiple benchmark datasets, providing a thorough assessment of various object detection methods. By extending the evaluation to datasets beyond the KITTI benchmark, this study offers a more holistic understanding of each approach's strengths and limitations. Additionally, we delve into persistent challenges in existing occlusion-handling approaches and emphasize the need for innovative strategies and future research directions to drive substantial progress in this field.

**Keywords:** video surveillance; object detection; deep learning; generative models; graphical models; data augmentation; occlusion handling





## 1. Introduction

The advent of smart cities has ushered in a transformative era in computer science, placing a significant emphasis on data acquisition and processing [1]. Within this landscape, cameras play a pivotal role by continuously capturing images and videos in various resolutions, ranging from HD to 4K and 8K [2]. This extensive data feed from cameras fuels computer vision and video surveillance applications, enabling the recognition of objects and events in both 2D and 3D realms [3]. These applications include diverse tasks, such as identifying and tracking suspicious objects, predicting hazardous scenarios, and facilitating real-time industrial analysis [4].

Object recognition and action/event detection serve as essential pillars of video surveillance systems, employing a combination of image-processing algorithms, convolutional morphological operations, and machine learning techniques to extract salient features and objects [5,6]. The recent advancements in high-speed computing and big data have propelled object detection systems to remarkable heights, achieving outstanding performance [7].

However, a significant challenge persists in these learning approaches—the handling of object occlusion as illustrated in Figure 1. Partial occlusion, especially in realistic environments where objects intertwine and obscure each other, introduces complexity into the data distribution. This intricacy involves overlaps in terms of shape, appearance, and positioning, making it challenging to describe using fixed training data [8,9].

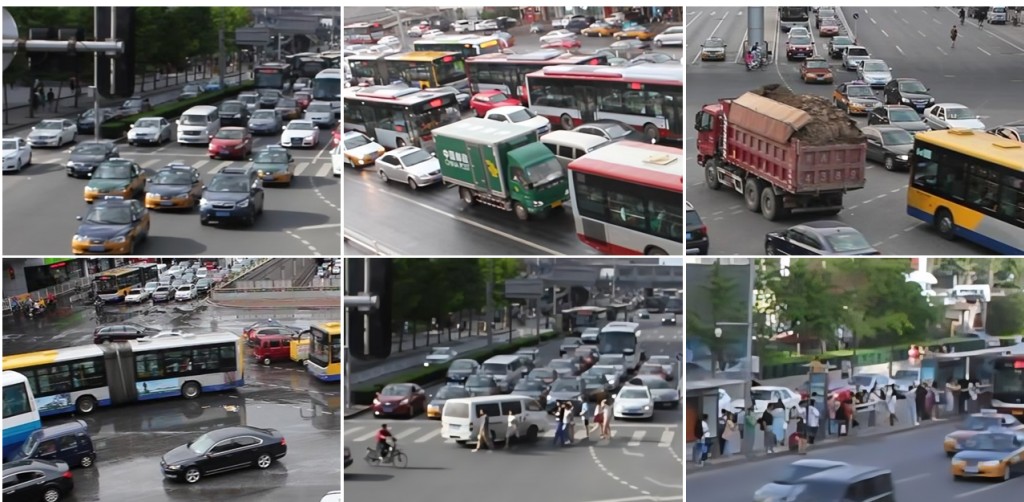

**Figure 1.** Challenging Object Detection Scenario: Illustration of a complex object detection scenario with elevated levels of partial occlusion in a crowded environment, utilizing the UA-DETRAC dataset [10].

Even with the state-of-the-art deep learning algorithms [11], there is a noticeable gap in emulating human-like recognition, particularly when faced with partially occluded objects [12]. This limitation becomes more apparent despite rigorous training on datasets with significant occlusion levels, indicating a fundamental deficiency in current computer vision systems. Research studies, such as that of Jha et al. [13], have highlighted the performance gap between modern deep neural networks and human vision in detecting partially occluded objects.

Motivated by the challenges posed by occlusion in object detection, this paper centers its attention on recent endeavors addressing object detection techniques tailored for occlusion handling (as illustrated in Figure 2). Occlusion handling involves developing algorithms and strategies that enable the accurate and robust detection of objects, even when they are partially obscured by other objects or background elements in the scene. The goal is to establish a cornerstone for forthcoming research in this field.

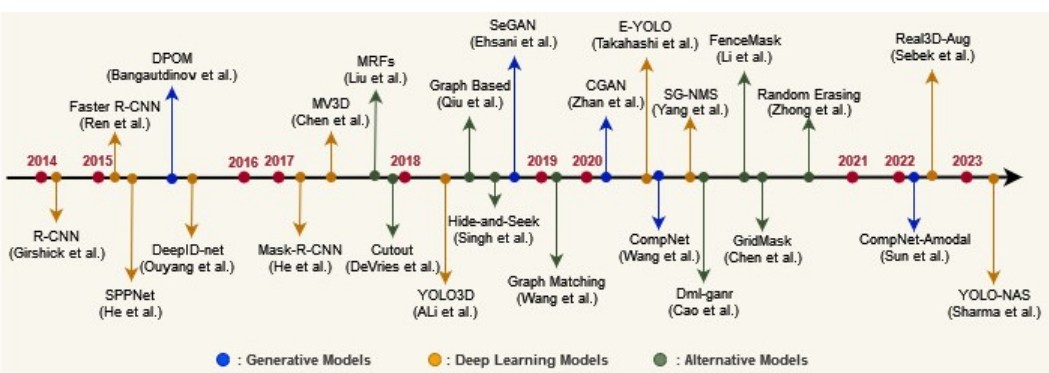

**Figure 2.** Evolution of some object detection approaches: a timeline of occlusion-handling techniques [14–33].

Previous recent surveys and reviews in the domain of occlusion handling have provided valuable insights into the evolving landscape of object detection. Notable works such as [34–37] have contributed to our understanding of various techniques and challenges associated with occlusion in object detection. However, to the best of our knowledge, our survey distinguishes itself by our distinctive contributions that comprise the following:

1.  **Thorough literature summary:** Provides a comprehensive overview of successful techniques for occlusion handling in object detection.

2.  **Extensive algorithm evaluation:** Conducts a detailed evaluation of current object detection algorithms in occlusion scenarios, offering experimental analysis and insights.
3.  **Identification of challenges:** Highlights the challenges faced by state-of-the-art deep learning algorithms in handling partial occlusion, emphasizing the need for more effective solutions.
4.  **Motivation for future research:** Serves as a foundational resource, guiding future research in the realm of object detection in occluded scenes.
5.  **Utilization of multi-object datasets:** Integrates the use of multi-object datasets for occlusion-handling evaluations, ensuring the robustness and applicability of proposed techniques to diverse real-world scenarios.

In summary, our contributions encompass a comprehensive overview, an extensive evaluation, experimental insights, exploration of alternative approaches, performance comparison of generative and deep learning models, identification of limitations, and guidance for future research.

The paper unfolds in two main sections. The Related Works section (Section 2) offers an insightful overview of effective techniques for occlusion handling, classifying them into distinct categories, including deep learning, generative models, graphical models, and data augmentation. Figure 3 explains the structural flow of mentioned methodologies. Following this, the Proposed Comparative Analysis section (Section 3) engages in a thorough examination of current object detection algorithms within occlusion scenarios, presenting detailed experimental analyses and valuable insights. The paper culminates with a comprehensive discussion in the Conclusion section, summarizing key findings, addressing limitations, and outlining potential avenues for future research.

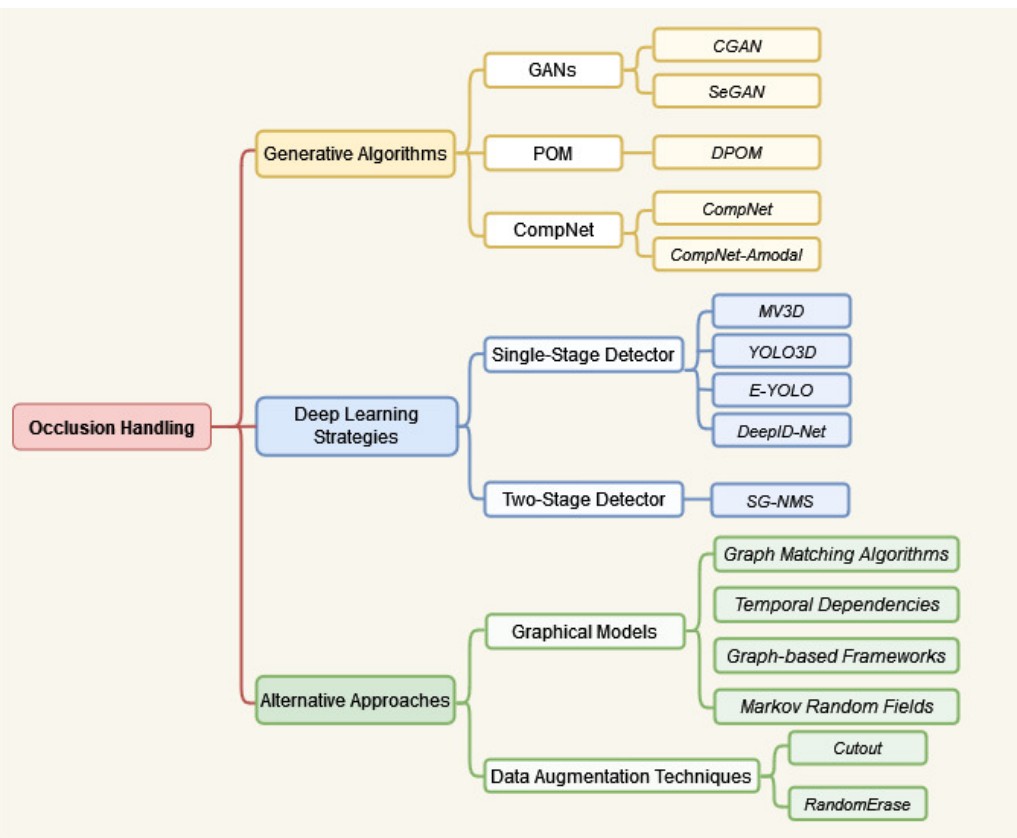

**Figure 3.** Taxonomy of occlusion-handling methodologies in object detection: an organized overview.

## 2. Related Works

One of the foremost challenges in object detection is the ability to accurately identify objects despite occlusion and deformations. Historical research in this domain has pre-

dominantly focused on two primary approaches: firstly, algorithms centered on extracting background information when objects are occluded and secondly, methods leveraging the depth information of the objects [14].

Early studies have shown that conventional learning algorithms tend to perform well when less than 10% of an object is occluded as reported by Perez et al. [38]. However, as the degree of occlusion increases, traditional methods encounter progressively higher detection failure rates, making it considerably challenging to recognize objects when occlusion levels reach approximately 50% [37]. In contrast, generative models have demonstrated remarkable capabilities in distinguishing between the background context and the targeted objects. This distinction significantly contributes to resolving occlusion-related issues [39]. Object detection has benefited greatly from state-of-the-art deep learning-based approaches, which we will delve into in Section 2.2.

In the context of our research, we categorize existing approaches into four distinct groups: (1) generative models, (2) deep learning approaches, (3) graphical models, and (4) data augmentation techniques. The generative models encompass methodologies that understand the generative process of objects, allowing for recognition of occluded object sections from various angles and spatial patterns. Conversely, deep learning approaches deploy neural networks to automatically learn object features. This automation not only reduces the time required for manual feature selection but also enhances detection speed and accuracy.

The quality, quantity, and dimensionality of the data, whether in the form of 2D or 3D images, exert a profound influence on model performance, especially during occlusion. Hence, the preprocessing and augmentation of data are crucial to address challenges associated with occlusion [14]. Additionally, graphical models provide a structured framework for modeling object relationships, incorporating contextual information and addressing occlusion challenges [40]. Data augmentation, as proposed by Cubuk et al. [41], enriches datasets with occluded objects and transformations, thereby enhancing the robustness and generalization of object detection models to occlusion scenarios.

In the subsequent subsections, we will delve into each of these four approaches, highlighting their methodologies and contributions to tackling the complex problem of occlusion in object detection.

### 2.1. Generative Algorithms for Occlusion Handling

Generative models, as applied to object detection under occlusion, play a crucial role in overcoming the challenges posed by obscured objects. These models employ a unique approach by simulating the generation process of occluded scenes, allowing them to distinguish between background context and targeted objects with a higher degree of accuracy [42]. Mathematically, generative algorithms aim to capture the intricate relationships between the observed data and the underlying occlusion patterns, enabling the synthesis of informative representations [43].

These generative models offer distinct advantages over conventional approaches, particularly when compared to training on occlusion-rich datasets or employing data augmentations [44]. By inherently understanding the generative process, these models excel in scenarios where objects are partially obscured. Unlike traditional methods, they do not solely rely on fixed training datasets, making them adaptable to varying occlusion levels, shapes, and appearances.

Generative models offer an innovative approach to addressing occlusion challenges by reconstructing obscured portions of objects. This section explores diverse generative models, shedding light on their distinct capabilities. Notable among these are Generative Adversarial Networks (GANs) [35] and, in the context of occlusion handling, the Probabilistic Occupancy Map (POM) method [45] and Compositional Generative Networks (CompNets) [15]. Each of these approaches contributes unique solutions to the complex issue of occlusion, providing a comprehensive overview of the various strategies employed in object detection.

2.1.1. Generative Adversarial Networks Approach

Generative Adversarial Networks (GANs) constitute a powerful unsupervised generative model (Figure 4) defined by a generator function ($G$) and a discriminator function ($D$) engaged in an adversarial learning process. The optimization objectives for the generator and discriminator in GANs are encapsulated in Equation (1) as initially designed by Goodfellow et al. [46]. The objective function $V(D, G)$ is defined as follows:

$$\min_G \max_D V(D, G) = \mathbb{E}_{x \sim p_{\text{data}}(x)}[\log D(x)] + \mathbb{E}_{z \sim p_z(z)}[\log(1 - D(G(z)))] \tag{1}$$

where $D$ distinguishes between real ($x$) and generated ($G(z)$) data, and $G$ creates data to deceive $D$. The first term encourages $D(x)$ to be close to 1 for real data, and the second term encourages $D(G(z))$ to be close to 0 for generated data. This adversarial game aims for an equilibrium where the generator produces data indistinguishable from real data, resulting in high-quality realistic samples.

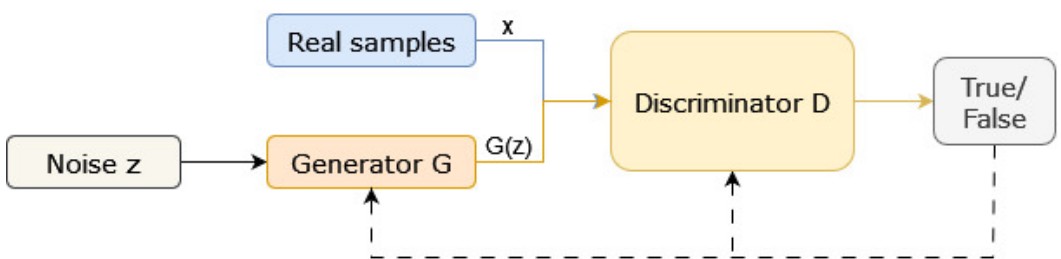

**Figure 4.** Computational workflow and architectural overview of Generative Adversarial Networks (GANs) [42].

Various GAN-based networks have demonstrated effectiveness in addressing occlusion challenges. For instance, Zhan et al. [16] employed Conditional GAN (CGAN) and partial convolution to regenerate the content of missing regions in a 2D image. This involved a self-supervised approach, where occluded data were labeled by strategically placing occluders from the dataset on objects. The model, composed of Partial Completion Network-mask (PCNet-M) and Partial Completion Network-content (PCNet-C), partially completed both the mask and appearance of the occluded object.

Similarly, Ehsani et al. [17] introduced SeGAN, a GAN-based model designed to generate the occluded regions of objects. SeGAN follows a two-step process: segmentation and painting. The segmentation part, a convolutional neural network (CNN), takes a 2D image and outputs a mask for the object based on information from visible regions. Subsequently, the painting part employs a conditional GAN to generate the occluded parts of the object.

Despite the notorious stability issues and training challenges of GANs, they prove instrumental in extending incomplete representations to complete ones [47]. However, their applicability varies across tasks; for instance, GANs perform exceptionally well in amodal appearance reconstruction but are less commonly employed in amodal segmentation and order recovery tasks. Combining GANs with other architectures and learning strategies enhances their potential across diverse occlusion-related challenges.

Nevertheless, GANs have their drawbacks, including stability issues and the complexity of training. While they exhibit impressive performance in amodal appearance reconstruction, their usage in amodal segmentation and order recovery tasks is comparatively limited [48]. Despite these challenges, leveraging GANs for occlusion handling remains pivotal in computer vision applications. Amodal completion tasks facilitated by GANs, such as predicting the full shape of objects and inferring occlusion relationships, contribute significantly to advancements in SVS applications. The creative and generative capabilities of GANs bring machine learning systems closer to human-like predictions of occluded areas, despite the existing challenges in their implementation.

### 2.1.2. Probabilistic Occupancy Map (POM) Approach

Probabilistic Occupancy Map (POM) [45] stands as a sophisticated multi-camera generative technique designed to deduce ground plane occupancy from diverse viewpoints through background removal. The methodology involves synthetically fitting a background subtraction model to binary foreground motion, allowing the repetitive computation of occupancy probabilities.

POM operates within a mathematical framework that excels in predicting ground plane occupancy at specific time steps. Unlike its counterparts, this framework boasts a distinct advantage; it enables the precise localization of individuals not only in camera views (RGB) but also on the 3D ground plane. This precision is achieved through the POM detector, which adeptly manages complex occlusion interactions among detected individuals. The detector employs an advanced generative model to estimate the probability of occupancy [49].

In an innovative extension, Timur et al. [18] build upon POM, introducing a model termed Depth Probabilistic Occupancy Map (DPOM). This model decodes images into a collection of people detection in crowded environments, utilizing POM as the foundational model for occlusion management. DPOM takes a step further by synthesizing depth maps instead of binary images. It explicitly considers occlusions while estimating the probability of target objects being present in the scene.

Experimental results using DPOM showcase an improvement in accuracy when leveraging object depth in the image. However, a trade-off arises, as DPOM modifies original images to extract object depth, resulting in information loss and hindering object distinguishability. This limitation negatively impacts the recognition phase, leading to the formation of unrecognizable objects [18].

### 2.1.3. Compositional Generative Networks Approach

Compositional Generative Network (CompNet) [15] stands as an innovative generative compositional model designed to accurately classify 2D images of partially occluded objects. Employing a voting system and an explicit representation of objects as parts, CompNet excels in the precise classification of objects based on the configuration of selected visible parts (Figure 5).

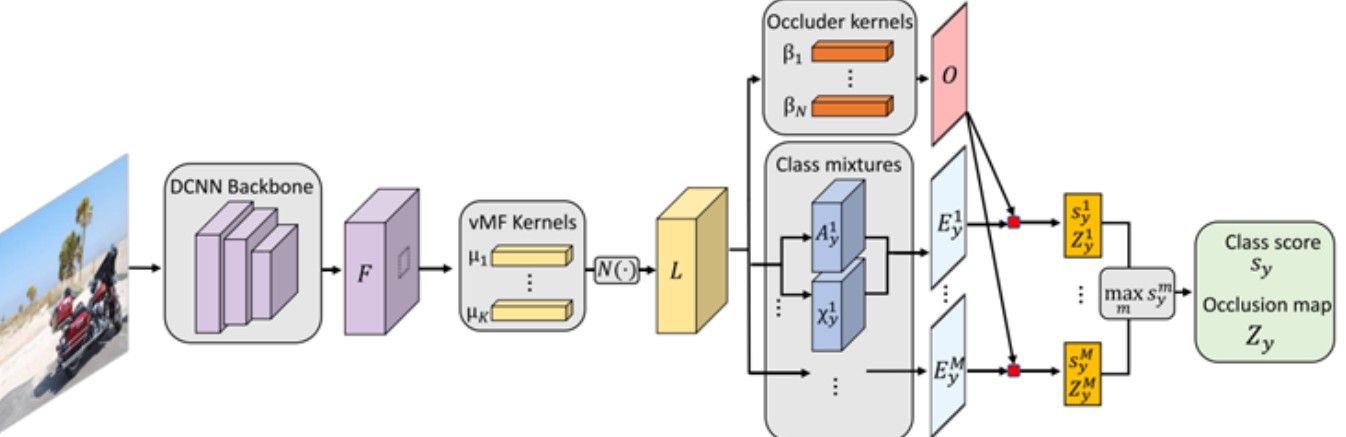

**Figure 5.** The CompNet classification model architecture: illustration depicting the feed-forward inference process within a CompNet for object classification [15].

However, similar to other Deep Convolutional Neural Network (DCNN) architectures, CompNet lacks the explicit disentanglement of context and object representations. Furthermore, it lacks robust mechanisms for estimating the bounding box of the object, rendering it unsuitable for object detection [15]. These limitations have been corroborated by Wang et al. [19], who demonstrated that biases in context, prevalent in training data, adversely

affect detection performance. Additionally, when objects are heavily occluded, the need to reduce detection thresholds increases, leading to an amplified influence of object context and resulting in false-positive detections in contextually barren regions.

To address these constraints, Wang et al. proposed significant enhancements to CompNet for the robust recognition of partially obscured objects [19]. They advocated deconstructing image representation into a mixture of context and object representations by generalizing contextual features through bounding box annotations in training data. Introducing a detection layer, they limited the impact of context on detection results. To enhance robust bounding box estimation, they expanded the CompNet part-based voting system, allowing votes for two opposing corners of the bounding box in addition to the object center.

In a related vein, Sun et al. [50] contributed to the research field of amodal segmentation under partial occlusion by inferring amodal segmentation into CompNet. Leveraging a Bayesian generative model with neural network features, they replaced the fully connected classifier in the CNN. The Bayesian model describes the image's features, including object classes and amodal segmentation, with a probability distribution. Although this method enhances the model's resistance to occlusion, its reliance on significant form priors limits its suitability to rigid objects, such as vehicles.

## 2.2. Deep Learning Strategies for Occlusion Handling

In recent years, deep learning has emerged as a powerful paradigm for tackling challenges posed by partial occlusion [51,52]. This section explores various deep learning-based strategies that have been employed to enhance object detection performance in the presence of occluded elements. Deep learning models have innovatively integrated concepts, such as part-based representations [53], refined decision processes [54], and the incorporation of 3D scene data [55], to leverage depth information effectively.

Among the various models in the deep learning realm, two prominent categories of CNN-based models have emerged for object detection [56]. The first category encompasses the two-stage detector method, exemplified by techniques like the Region-Convolutional Neural Network (R-CNN) series [20] and Spatial Pyramid Pooling Networks (SPPNets) [57]. In these methods, the detection process involves segregating target location and recognition into distinct components. However, the computational expense and relatively slower detection speed have been drawbacks of two-stage frameworks.

On the other hand, the second category presents single-stage frameworks that sacrifice some accuracy for enhanced speed. Methods like the You Only Look Once (YOLO) series [58] (including YOLOv5 [59], YOLOv6 [60], YOLOv7 [61], YOLOv8 [62], and YOLO-NAS [63]), Single-Shot MultiBox Detector (SSD) [64], and OverFeat [65] fall into this category. Single-stage frameworks bypass the region proposal generation stage, directly predicting class probabilities and bounding box offsets from entire images simultaneously.

It is imperative to note that while these exemplary networks have significantly contributed to object detection, they were not explicitly developed to handle occlusion challenges. As such, their performance with occlusion is limited. The subsequent sections (Sections 2.2.1 and 2.2.2) will delve into models specifically tailored for SVSs, shedding light on how they navigate the unique challenges posed by occlusion in this context.

### 2.2.1. Single-Stage Detector Algorithms

Chen et al. [14] introduced an innovative approach to address occlusion challenges by leveraging precise depth information obtained from 3D image sensors, specifically Laser Imaging Detection and Ranging (LIDAR). Their proposed model, the end-to-end multi-view 3D object detection network (MV3D), integrates a regional fusion network and a 3D proposition network. Operating on both the bird's eye view and frontal views of the LIDAR point cloud, along with an RGB image, MV3D capitalizes on the complementary strengths of the accurate depth data of LIDAR and the rich visual features from the camera.

By adopting the "LIDAR–camera fusion" paradigm, which harnesses the strengths of both LIDAR and RGB image inputs, MV3D optimally utilizes the accurate depth information provided by LIDAR and the rich visual features captured by the camera. This synergistic combination results in a more precise object detection and recognition process. MV3D initiates the pipeline by generating 3D object proposals from a bird's-eye-view map, projecting them into three distinct views. The deep fusion network then integrates region-wise features through an ROI pooling layer for each view. This fused information becomes instrumental in predicting object classes and executing oriented 3D box regression. The ultimate output of MV3D consists of oriented 3D bounding boxes as visually depicted in Figure 6.

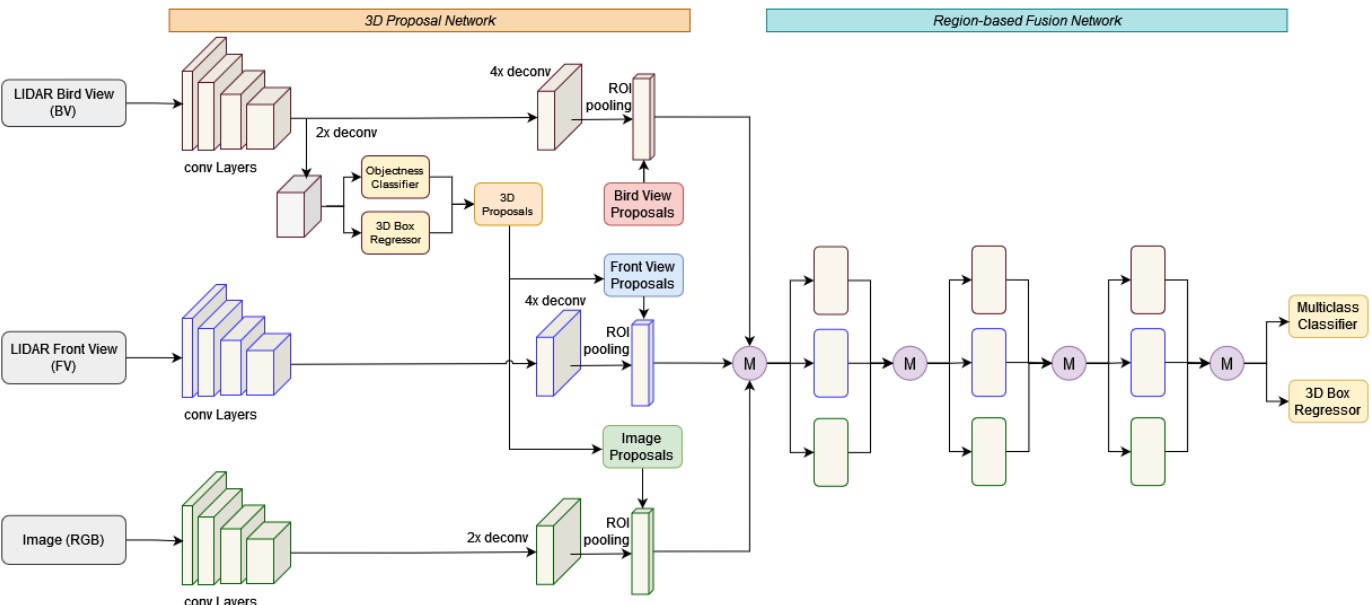

**Figure 6.** MV3D Architecture: Overview of the multi-view 3D object detection network, highlighting the fusion of LIDAR and RGB information for enhanced object detection in 3D space [14].

A notable advantage of this model is its capability to handle occlusions. By directly performing 3D object detection on the truncated cone corresponding to the 2D bounding box in the RGB image, MV3D enhances accuracy in the presence of occluded objects. However, the fusion of LIDAR and camera data introduces additional time complexity to the overall detection process, impacting real-time applications.

Ali et al. [21] introduced a distinctive solution, YOLO3D, an extension of the YOLOv4 generic object detector [66], specifically designed for handling occlusion scenarios. YOLO3D operates on the "LIDAR-only" paradigm, utilizing the projected LIDAR point cloud as a bird's-eye-view grid to preserve essential 3D information. In contrast to MV3D, YOLO3D focuses solely on LIDAR data for object detection. YOLO3D extends the successful single-shot regression meta-architecture from 2D perspective images to generate oriented 3D object bounding boxes. This approach provides a unique perspective on handling occlusion by emphasizing the depth information captured by LIDAR. However, it is essential to note that the detection accuracy of YOLO3D falls short when compared to MV3D. The choice between "LIDAR-only" and "LIDAR–camera fusion" paradigms may depend on specific use cases and the trade-off between simplicity and detection performance.

Takahashi et al. [22] introduced the expandable YOLO (E-YOLO) technique, an enhanced version of YOLOv3 [67] designed to address occlusion challenges effectively. Operating under the "Camera-only" paradigm, E-YOLO (Figure 7) leverages the strengths of a stereo camera to achieve high-quality 3D object recognition, particularly during occlusion scenarios.

The model utilizes edge detection and frame differences to enhance the detection process, incorporating prior knowledge about the size of 2D objects. By predicting 3D bounding boxes based on stereo camera information, E-YOLO perfoms well when handling occluded objects. Experimental results highlight the algorithm's ability to overcome limitations associated with the frame difference method, resulting in a high-speed detection characteristic with promising commercial applications.

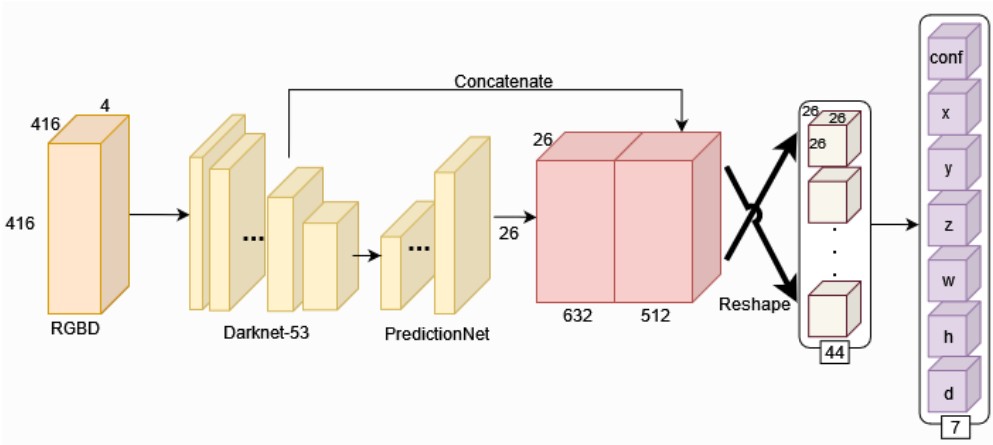

**Figure 7.** E-YOLO framework architecture: illustration of the Expandable YOLO (E-YOLO) framework, emphasizing the integration of edge detection and frame differences for improved 3D object recognition using a stereo camera [22].

However, it is important to note that E-YOLO exhibits lower discrimination in detecting non-occluded objects compared to DCNNs. The trade-off between speed and discrimination capabilities should be considered based on specific application requirements and priorities.

Wanli et al. introduced DeepID-Net [23], a deformable deep convolutional neural network (DCNN) tailored for general object detection. This network incorporates a deformation-constrained pooling layer designed to consider part deformation during object detection. Inspired by deformable RoI pooling, this layer significantly enhances the model's ability to handle complex object structures.

Despite its increased complexity, DeepID-Net utilizes an innovative pre-training technique that enhances the efficacy of model averaging. This technique enables the network to learn feature representations better suited for the object identification problem. The proposed algorithm further suggests that various components interact automatically through jointly learned deep features, deformable parts, occlusion handling, and classification processes. This holistic approach contributes to the network's robust performance in addressing occlusion challenges during object detection.

### 2.2.2. Two-Stage Detector Algorithms

Yang et al. [24] introduced the Semantics-Geometry Non-Maximum-Suppression (SG-NMS) algorithm as part of the Serial R-FCN network [68], aiming to enhance object detection through a heuristic-based approach. This two-stage detector combines bounding boxes based on detection scores obtained from the Serial R-FCN, suppressing overlapping boxes with lower scores.

The processing pipeline begins with a backbone CNN that analyzes the input image, producing feature maps (Figure 8). A Region Proposal Network (RPN) [25] identifies regions of interest (ROIs). In the serial pipeline, the regression head refines ROIs before passing them to the classification head, generating detection scores. An additional Semantics–Geometry Module, unique to the SG-NMS algorithm, learns Semantics–Geometry Embeddings (SGEs) for each refined box. The SG-NMS algorithm utilizes these embeddings, detection scores, and detected boxes to determine the final detections.

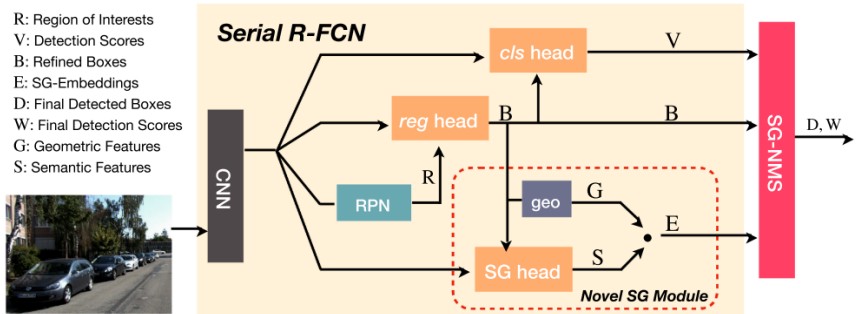

**Figure 8.** SG-Det model overview: A concise depiction illustrating the essential components and structure of the SG-Det model. This model incorporates SG-NMS to improve detection performance, especially in scenarios involving overlapping bounding boxes and occluded objects [24].

SG-NMS addresses occlusion challenges by mapping potential detections to a latent space, dividing occluded objects in the image. Detections belonging to the same physical object cluster tightly in this space, while those belonging to different objects are pushed apart. The algorithm selects the box with the highest detection score as the pivot box for the SGE, measuring distances to determine box retention. If the distance exceeds a predefined monotonically increasing function, the box is retained. This approach effectively manages occlusion scenarios during object detection.

### 2.3. Alternative Approaches for Occlusion Handling

While generative models and deep learning strategies contribute significantly to occlusion handling, alternative approaches offer diverse perspectives and techniques to enhance object detection performance in challenging scenarios. In this section, we delve into alternative strategies that go beyond generative models and deep learning, exploring the realms of graphical models [34] and data augmentation [41].

#### 2.3.1. Graphical Models for Occlusion Handling

In the realm of addressing occlusion challenges in object detection, graphical models offer a distinctive approach by leveraging probabilistic relationships and structural dependencies. These models, such as graph-matching algorithms [26] and probabilistic graphical models [69], provide a powerful framework for enhancing data association and object detection performance in complex scenarios with occlusion.

Graph-matching algorithms play a pivotal role in establishing correspondences between nodes in different graphs, where each node represents an object or a feature. This approach formulates data association as an optimization problem, with quadratic graph matching being a notable technique [70]. It minimizes pairwise dissimilarity measures between nodes, leading to improved object detection accuracy. Noteworthy applications include frameworks combining quadratic graph matching with deep learning features, showcasing significant enhancements, particularly under challenging conditions like occlusions.

The integration of temporal dependencies becomes crucial in scenarios where objects undergo occlusion or abrupt changes. Techniques such as temporal–spatial structured deep metric learning utilize recurrent neural networks (RNNs) [71] to capture both short-term and long-term temporal dependencies [27]. This approach enhances object detection performance, especially in the presence of occlusions. However, challenges may arise in scenarios with highly complex motion patterns.

Graph-based frameworks, incorporating spatial constraints, provide a structured way to represent relationships between objects across frames. Objects are depicted as nodes, and pairwise spatial relationships define edge weights. This approach enforces consistent associations based on spatial relationships, resulting in improved object detection accuracy [28]. While effective, challenges may emerge in environments with dynamically changing spatial relationships.

Probabilistic graphical models, such as Markov Random Fields (MRFs), represent a powerful tool for modeling complex dependencies and incorporating contextual information in object detection [29]. MRFs utilize an undirected graph to represent joint probability distributions over variables. The incorporation of appearance and motion cues, along with contextual information, through MRFs enhances data association in object detection, particularly in challenging scenarios with occlusions.

The exploration of graphical models in occlusion handling provides valuable insights into their strengths and limitations [27–29,70]. From quadratic graph matching to the utilization of temporal–spatial dependencies, these models contribute to advancing the field of object detection under complex scenarios, shedding light on tailored solutions for robust data association.

### 2.3.2. Data Augmentation for Occlusion Handling

Beyond generative models, data augmentation emerges as a pivotal strategy to bolster the resilience of object detection models when confronted with the complexities of occlusion [44]. This approach is centered on enriching datasets with a spectrum of occlusion scenarios, thereby equipping the model to navigate variations in object visibility with greater adeptness.

Within the realm of data augmentation, region-level strategies [72] have risen to prominence due to their effectiveness, particularly in domains such as object detection within crowded settings or visual tracking, where occlusion presents formidable challenges. Diverging from the traditional approach of uniformly transforming entire input images, these strategies tactically apply image transformation techniques to localized patches. The essence of region-level augmentation lies in its ability to forestall overfitting to specific features within localized sub-regions of training samples, accomplished by infusing diversity through targeted transformations.

In addition to Cutout and Random Erasing, various other techniques like Hide and Seek [30] , FenceMask [31], GridMask [73], and GridCut are displayed in Figure 9, showcasing the diversity of region-level data augmentation strategies.

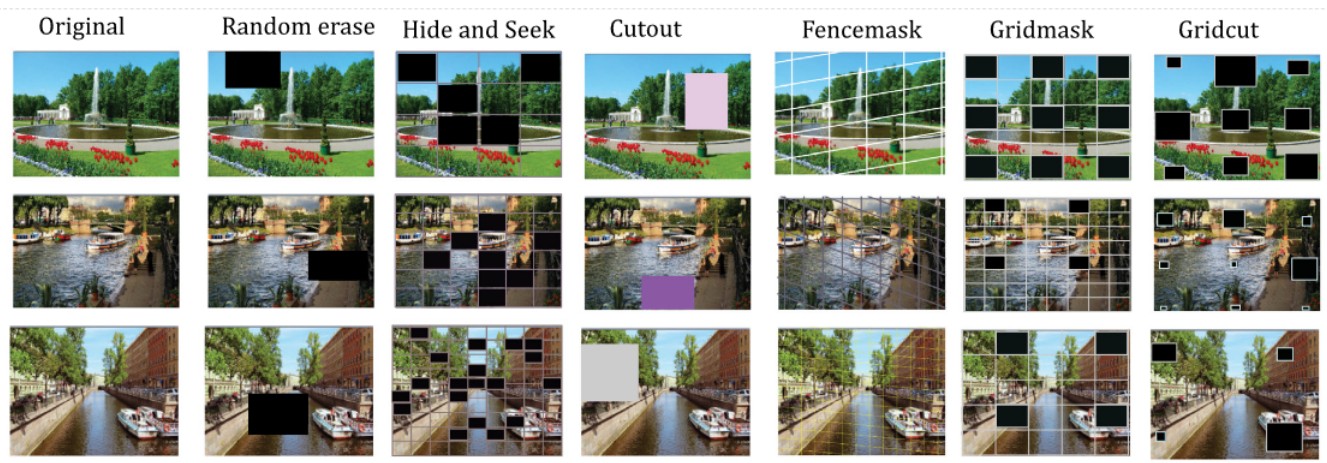

**Figure 9.** Visual effects of various region deletion methods for enhanced occlusion handling in object detection [44].

Region deletion, akin to dropout regularization but at the input level, involves eliminating random regions from input samples during training. This technique, exemplified by approaches like Cutout [32] and Random Erasing [33], contributes to the introduction of diversity by varying visual features across different training samples. In the context of occlusion simulation, region deletion proves particularly valuable. Visual tracking applications, where occlusion is commonplace due to object interactions or dynamic environmental changes, benefit from this approach.

While region-level deletion strategies offer benefits in simulating occlusion and preventing overfitting, certain limitations exist. The effectiveness of these approaches may vary based on the specific characteristics of the dataset and the nature of occlusion scenarios [44]. Additionally, careful consideration is required when setting hyperparameters, such as deletion probabilities and aspect ratios, to ensure that the augmentation process remains realistic and aligns with the challenges posed by occluded objects.

*2.4. Summary*

In navigating the challenges of occlusion in object detection, we explored a diverse array of approaches, encompassing generative models (including GANs, POM, and CompNets), deep learning approaches, graphical models (encompassing graph-matching algorithms, temporal dependencies, graph-based frameworks, and Markov random fields), and data augmentation through region-level strategies. Deep learning techniques delve into the intricate layers of neural networks, leveraging diverse strategies like depth extraction and penalty-based mechanisms to enhance object detection in the face of occlusion. Generative models offer unique perspectives, distinguishing background context from target objects and utilizing innovative approaches like CompNet. Graphical models provide robust solutions, incorporating high-order structure features and probabilistic frameworks to model complex dependencies and improve data association in multiple object tracking. Region-level data augmentation strategies, a subset of data augmentation, focus on localized patches, providing effective tools such as region deletion to simulate occlusion conditions and prevent overfitting. Although each approach helps to address the challenges of occlusion, a comprehensive comparative analysis is crucial for discerning their strengths, limitations, and overall effectiveness. The upcoming Section 3 will illuminate the nuances through experimental evaluations, offering insights into the performance of selected object detection models, ultimately bridging the existing gap between automated detectors and human-level perceptual capabilities.

**3. Proposed Comparative Analysis**

*3.1. Occlusion-Handling Databases*

Several datasets play a pivotal role in evaluating occlusion-handling approaches, each offering unique challenges and diverse scenarios for comprehensive testing. The Occluded-Vehicles dataset, derived from the PASCAL3D+ dataset by Xiang et al. [74], artificially introduces occlusions using segmented objects, white patches, random noise, and textures, providing distinct occlusion levels (L0 to L3). Kortylewski et al. [39] present the Occluded-COCO-Vehicles dataset, featuring real-world occlusions by removing objects from the MS-COCO dataset, aligning with the Occluded-Vehicles' occlusion levels.

The choice of datasets was deliberate, with each serving a specific purpose in our study. The KITTI dataset [75], developed jointly by the Karlsruhe Institute of Technology and the Toyota Technological Institute at Chicago, offers 2D and 3D images, accommodating a wide range of occlusion complexities. Additionally, the NuScenes dataset by Caesar et al. [76] focuses on autonomous driving scenarios, PascalVOC 2012 [77] extends the PASCAL3D+ dataset for robust object detection, and CityPersons by Zhang et al. [78] addresses pedestrian detection in urban environments.

The use of these databases is outlined in Table 1. It is essential to note that each dataset served a specific purpose, contributing to a more comprehensive evaluation of occlusion-handling models. The rich set of challenges presented by these datasets encompasses diverse occlusion scenarios and real-world complexities, ensuring a thorough examination of occlusion-handling models tailored to different scenarios.

**Table 1.** Summary of 3D and 2D datasets for multi-object detection occlusion analysis.

| Dataset | Description | Number of Classes | Number of Images | Data Type |
|---|---|---|---|---|
| KITTI [75] | Real-world urban scenes with varying occlusion levels (high occlusion focus) | 3 (Car, Pedestrian, Cyclist) | 7481 | 3D LIDAR scans and camera images |
| NuScenes [76] | Diverse urban driving scenarios with extensive sensor data (low occlusion focus) | 10+ | 1000+ | 3D LIDAR scans and camera images |
| OccludedPascal3D [74] | Varied indoor and outdoor scenes with varying occlusions (high occlusion focus) | 12 | 2073 | RGB Images and point clouds |
| PascalVOC 2012 [77] | Diverse scenes and environments for object detection (low occlusion focus) | 20+ | 17,125 | Camera images |
| CityPersons [78] | Urban pedestrian detection with varying occlusion levels (high occlusion focus) | 1 (Pedestrian) | 5000 | Camera images |

### 3.2. Evaluation Criteria

The KITTI dataset is a comprehensive benchmark for object detection and 3D orientation estimation. Its evaluation criteria for occlusion handling involve precise 2D/3D bounding boxes for object classes like cars, vans, trucks, pedestrians, cyclists, and trams. Occlusion levels are categorized into three complexity levels: "Easy" (objects fully visible), "Moderate" (partial occlusion), and "Difficult" (objects challenging to identify) [75]. The dataset adopts principles like "DontCare" annotation for areas with distant or occluded objects, not counting them as true positives (TPs) or false positives (FPs). False positives and false negatives are treated differently based on the overlap relative to the predicted bounding box's area. Objects with a height of less than 30 pixels are excluded from evaluation due to their susceptibility to error [79].

The CityPersons dataset focuses on pedestrian detection in urban environments, and its evaluation criteria for occlusion handling consider annotated objects under different occlusion scenarios. Occlusion levels are specified, and evaluations are conducted separately for each level, allowing a detailed analysis of the model's performance in handling occluded instances. False positives and false negatives are carefully addressed, with specific rules in place for handling occluded regions [78]. Similar to KITTI, difficulty levels are defined based on factors like occlusion, and evaluations are reported independently for different difficulty levels. The dataset's evaluation criteria provide a robust framework for assessing pedestrian detection performance under various occlusion conditions in urban settings.

The PASCAL VOC 2012 dataset is a widely used benchmark for object detection, and its evaluation criteria for occlusion handling involve annotated objects with distinct occlusion levels. These levels typically include "unoccluded", "partially occluded", and "heavily occluded". Evaluation metrics encompass precision, recall, and the F1 score, reported separately for different occlusion levels. False positives and false negatives are carefully handled, considering specific rules for occluded regions [77]. The dataset categorizes objects into difficulty levels based on occlusion, size, and truncation, and evaluations are reported independently for each difficulty level. This class-wise evaluation provides insights into how well a model handles occlusion across different object categories.

### 3.3. Experimental Results

For the sake of interpretability, researchers often utilize average precision (AP) as a standard metric. AP is considered superior to the F1 score, primarily because it provides a comprehensive measure across various thresholds. The calculation of this metric involves

the cumulative distribution of true positives (TPs), false positives (FPs), and false negatives (FNs) as expressed in the following formulas:

$$AP = \sum_{k=0}^{k=n-1} [Recalls(k) - Recalls(k+1)] * Precisions(k) \tag{2}$$

With : $Recalls(n) = 0$, $Precisions(n) = 1$, $n = Number\ of\ thresholds$
And :

$$Recall = \frac{tp}{tp+fn} \tag{3}$$

$$Precision = \frac{tp}{tp+fp} \tag{4}$$

Here, $Recalls(k)$ denotes the recall at a specific threshold $k$, signifying the ratio of true positives correctly identified among all actual positives. Correspondingly, $Precisions(k)$ represents precision at the same threshold, indicating the ratio of true positives among all instances predicted as positives. These values are inherently threshold-dependent, with $k$ representing various threshold levels. The calculation of AP, as expressed in Equation (2), involves a comprehensive assessment across different thresholds, capturing the nuanced performance of the object detection model [80].

Our experimental design involves three main comparisons. Initially, we evaluate established deep learning networks on KITTI 2D and CityPersons datasets, focusing on their performance in handling occlusion. Subsequently, we explore alternative occlusion-handling approaches, such as YOLONAS-Cutout [32], DeepLab-CRF [81], and RGRN [82], using the Pascal VOC 2012 dataset. Finally, we conduct a comparative analysis between generative (CompNet [15]) and specialized deep learning models (MV3D [14] and YOLO3D [21]) designed explicitly for occlusion scenarios. We use the AP metric for interpretability and present diverse datasets to assess model robustness across varying complexities and occlusion levels.

In our comprehensive experiments, we rigorously assessed the performance of established deep learning networks on KITTI 2D and CityPersons datasets, with a specific emphasis on addressing challenges related to occlusion. As detailed in Table 2, the AP values for Car, Pedestrian, Cyclist (KITTI 2D), and Person (CityPersons) reveal crucial insights. Notably, during low-occlusion conditions, all models demonstrate competitive AP values, with YOLO-NAS consistently outperforming others as showcased by its top-ranking AP values. However, when confronted with high occlusion levels during testing on both KITTI and CityPersons, a decline in AP is observed across all models. Specifically, YOLO-NAS maintains the highest performance, but the decrease in AP highlights the inherent challenges faced by 2D object detection models in real-world scenarios with significant occlusion. The nuanced analysis of AP values for each class provides a detailed understanding of the models' strengths and weaknesses, emphasizing the need for robust occlusion-handling strategies in future developments (see Figure 10).

**Table 2.** Object detection AP results for deep learning models without occlusion handling on KITTI 2D [79] and CityPersons [78] datasets under high-occlusion conditions.

| Model | AP(%) (KITTI 2D) | | | (CityPersons) |
|---|---|---|---|---|
| | Car | Pedestrian | Cyclist | Person |
| F-RCNN | 57.2 | 53.8 | 48.5 | 79.3 |
| YOLOv5s | 65.6 | 63.3 | 58.4 | 82.5 |
| YOLOv6s | 66.7 | 64.4 | 58.1 | 84.1 |
| YOLOv7 | 59.2 | 58.4 | 47.6 | 80.9 |
| YOLOv8s | 67.2 | 63.4 | 60.7 | 85.7 |
| **YOLO-NAS** | **69.2** | **64.3** | **61.7** | **87.3** |

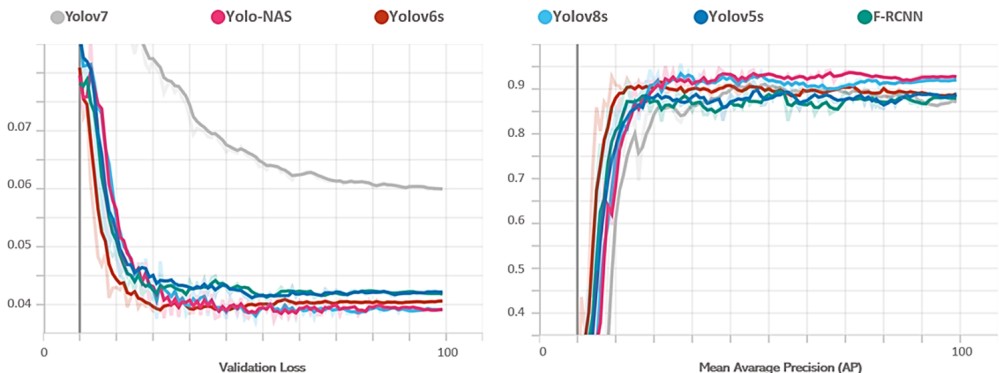

**Figure 10.** Training progress: average precision and validation loss for deep learning networks without occlusion handling.

Turning our attention to the exploration of alternative approaches for occlusion handling, we conducted a meticulous evaluation of a modified version of YOLO-NAS, incorporating both the Cutout and GridMask techniques. The comparison also included graphical models, DeepLab-CRF, and RGRN, with a focus on improving analytical functions in image processing. The results, as presented in Table 3, reveal compelling insights into the performance of these techniques across various object classes. Notably, the YOLO-NAS model enhanced with the Cutout technique exhibits superior performance, achieving the highest AP values across almost all object classes. This outcome signifies the effectiveness of the Cutout approach in mitigating the impact of occlusion on object detection. DeepLab-CRF demonstrates competitive results, outperforming RGRN in several classes. The nuanced analysis of AP values per class provides valuable information for researchers and practitioners seeking effective occlusion-handling strategies in real-world scenarios.

**Table 3.** Alternative approaches for object detection AP results on PascalVOC dataset [77].

| Model | AP per Class | | | | | | | | | | | | | | | | | |
|---|---|---|---|---|---|---|---|---|---|---|---|---|---|---|---|---|---|---|
| | *aeroplane* | *bird* | *bicycle* | *boat* | *bottle* | *bus* | *car* | *cat* | *chair* | *cow* | *dining table* | *dog* | *horse* | *motorbike* | *person* | *potted plant* | *sheep* | *sofa* |
| DeepLab-CRF | 79.3 | 76.7 | 78.7 | 77.6 | 76.1 | 78.5 | 79.5 | 74.8 | 76.4 | 73.7 | 78.4 | 73.7 | 78.0 | 78.9 | 79.1 | 76.8 | 74.4 | 77.3 |
| RGRN | 76.3 | 73.7 | 75.9 | 74.3 | 72.6 | 75.9 | 76.9 | 71.8 | 73.6 | 71.0 | 76.1 | 70.9 | 75.6 | 75.9 | 76.2 | 73.9 | 70.7 | 75.1 |
| **YOLONAS-Cutout** | **92.4** | **88.7** | **89.8** | **88.3** | **87.9** | **90.1** | **91.3** | **86.6** | **88.3** | 85.8 | **91.2** | 85.9 | **91.0** | **89.3** | **91.4** | **87.8** | **86.5** | **88.6** |
| YOLONAS-GridMask | 89.1 | 86.1 | 87.6 | 86.4 | 85.4 | 88 .3 | 89.2 | 83.5 | 85.9 | 82.9 | 89.2 | 82.9 | 89.0 | 87.0 | 89.3 | 85.5 | 83.5 | 87.0 |

In the culminating experiment dedicated to occlusion handling for SVSs, we conducted an exhaustive comparison using the KITTI dataset. This evaluation focused on assessing the performance of various approaches across different occlusion levels, providing nuanced insights into their efficacy. The results, outlined in Table 4, showcase distinct strengths among the evaluated networks, including both occlusion-handling object detection methods and alternative approaches for object detection referenced in another table, ensuring a comprehensive analysis using the same dataset for a more holistic perspective.

MV3D emerges as the standout performer, particularly excelling in scenarios with 'Moderate' and 'Hard' occlusion levels, demonstrating superior efficacy with AP values of 88.4% and 86.1%, respectively. CompNet also demonstrates commendable performance across different occlusion levels, exhibiting competitive results, with AP values ranging from 72.3% to 81.6%. YOLO3D, designed for LIDAR data, exhibits relatively lower efficacy, particularly in challenging scenarios with increased occlusion, presenting AP values ranging from 49.3% to 79.8%. Notably, the inclusion of YOLONAS-Cutout enriches our

understanding by revealing its proficiency in scenarios with 'Easy' occlusion levels, with precision gradually decreasing as the occlusion levels intensify as indicated by the AP values. The network comparison indicates that YOLONAS-Cutout performs exceptionally well in detecting objects in less challenging scenarios. Furthermore, our evaluation of CompNet involved adapting its original object segmentation [50] evaluation script into a detection network, emphasizing its versatility. These comprehensive results offer valuable insights into the relative strengths of generative and deep learning approaches tailored for SVS applications.

**Table 4.** AP results for occlusion-handling approaches on KITTI dataset [79].

| Network | Data | Car | | | Pedestrian | | | Cyclist | | |
|---------|------|-----|-----|-----|-----|-----|-----|-----|-----|-----|
| | | **Easy** | **Moderate** | **Hard** | **Easy** | **Moderate** | **Hard** | **Easy** | **Moderate** | **Hard** |
| YOLO-NAS | 2D | 97.4 | 80.5 | 69.2 | 92.5 | 76.8 | 64.3 | 89.8 | 73.2 | 61.7 |
| DeepLab-CRF | 2D | 79.3 | 70.7 | 68.7 | 76.1 | 69.5 | 65.5 | 74.8 | 66.4 | 63.7 |
| RGRN | 2D | 76.3 | 69.7 | 67.9 | 74.3 | 67.6 | 63.9 | 71.8 | 64.6 | 60.0 |
| YOLONAS-Cutout | 2D | **98.0** | 85.5 | 73.2 | **94.7** | 80.2 | 69.9 | **90.9** | 75.1 | 70.7 |
| YOLONAS-GridMask | 2D | 97.5 | 82.3 | 70.8 | 93.1 | 79.6 | 68.2 | 89.3 | 74.8 | 69.1 |
| CompNet | 2D | 81.6 | 76.8 | 72.3 | 78.9 | 71.2 | 66.7 | 75.2 | 69.1 | 65.6 |
| YOLO3D | LIDAR | 79.8 | 64.5 | 49.3 | 75.2 | 60.1 | 45.8 | 69.7 | 54.6 | 39.4 |
| MV3D | 2D + LIDAR | 92.5 | **88.4** | **86.1** | 89.7 | **84.5** | **81.2** | 82.6 | **79.1** | **75.4** |

*3.4. Discussion*

In our experimental evaluation, we observed nuanced performance results in various occlusion-handling approaches, revealing challenges for deep learning networks, particularly in scenarios with significant occlusion. Alternative strategies and the impact of data augmentation techniques on precision and recall, especially in the presence of occluded objects, played crucial roles. Tailoring occlusion-handling models for SVS applications, specifically with the KITTI dataset, exposed distinct performance variations. MV3D demonstrated superiority in moderate and hard occlusion levels, while CompNet exhibited commendable performance. However, YOLO3D showed reduced efficacy as the occlusion levels intensified. Acknowledging challenges in severe occlusion scenarios, dynamic changes, and diverse object scales, we underscore the ongoing need for research to address these complexities. Looking ahead, our future directions emphasize refining existing models, exploring hybrid approaches, and proposing innovative evaluation metrics to advance the field's understanding and capabilities. This comprehensive analysis provides a deeper understanding of occlusion handling in object detection within the context of Surveillance Video Systems.

## 4. Conclusions and Future Directions

In conclusion, our survey meticulously explored diverse strategies for occlusion handling in object detection, categorizing them into distinct approaches such as deep learning, generative models, graphical models, and data augmentation. Our experiments, particularly within the realm of SVS using the KITTI dataset, shed light on the relative strengths and limitations of various occlusion-handling techniques. While deep learning models exhibit remarkable performance, challenges persist, especially in the face of significant occlusion. The comparative analysis, including alternative approaches and specialized models, emphasized the need for context-specific strategies. Recognizing the significance of data augmentation in enhancing occlusion handling, our findings highlight the nuanced performance of different models across varying occlusion levels.

However, it is crucial to acknowledge inherent limitations, including challenges in severe occlusion scenarios, dynamic changes, and diverse object scales, which impact precision and recall. Adapting object detection models for SVS applications requires a delicate balance between accuracy and speed, particularly in real-time scenarios. These

identified limitations underscore the necessity for ongoing research and innovation in occlusion-aware object detection.

Our future research endeavors will be anchored in addressing the concrete challenges identified during our study. To enhance the effectiveness of occlusion-handling frameworks in SVS applications, we will concentrate on refining existing models, exploring hybrid approaches, and advancing evaluation metrics. Specifically, we aim to make significant contributions to the field by focusing on the detection of objects and handling occlusion primarily from 2D images. Building upon the limitations observed in current approaches, our work will strategically leverage existing models that extract depth from 2D images. Through enhancements and tailored modifications, we anticipate providing a more nuanced understanding of occlusion patterns. This strategic approach is poised to contribute substantially to the development of robust object detection frameworks capable of addressing occlusion challenges more effectively.

In summary, this survey, coupled with practical experiments, serves as a comprehensive guide for future research and the development of more resilient frameworks for occlusion-aware object detection, contributing to the evolving landscape of computer vision applications.

**Funding:** This research received financial support from ARES as part of a Ph.D. program conducted through joint supervision between UMONS in Belgium and UM5 in Morocco.

**Data Availability Statement:** Datasets analyzed during the current study are available in the following website links: https://www.cvlibs.net/datasets/kitti/eval_3dobject.php (KITTI Vision Benchmark), https://paperswithcode.com/dataset/pascal-voc (Pascal Voc 2012), https://www.nuscenes.org/ (Nuscenes), https://paperswithcode.com/dataset/citypersons (CityPersons) and https://paperswithcode.com/dataset/pascal3d-2 (Occluded Pascal 3D+) (accessed on 6 December 2023).

**Conflicts of Interest:** The authors declare no conflicts of interest.

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
