# Peer review of "Enhancing Object Detection in Smart Video Surveillance: A Survey of Occlusion-Handling Approaches"

_electronics, doi:10.3390/electronics13030541_

Round 1

Reviewer 1 Report

Comments and Suggestions for Authors

This paper reviewed the developed occlusion handling approaches in object detection involving the generative algorithms, the deep learning strategies and alternative approaches. The descriptions of these approaches were written very well. Authors are recommended to consider the following points listed below further.

1. In experiments, authors evaluated established deep learning networks on KITTI 2D and CityPersons datasets, explored alternative occlusion-handling approaches using the Pascal VOC 2012 dataset, and conducted analysis between generative and some deep learning models.In section 3.1, authors introduced NuScenes and OccludedPascal3D datasets besides KITTI , CityPersons and Pascal VOC 2012 datasets. Why not evaluate the selected approaches by using a same dataset? Why did you neglect NuScenes and OccludedPascal3D datasets in experiments?

2. In section 3.3, only the average precision (AP) was used to evaluate the experimental results. Please provide other metrics such as mAP, Recall and Precision in order to compare the approaches more completely.

Author Response

Greetings,

I trust this message finds you well, and I want to express our gratitude for your insightful review of our manuscript. Your feedback has been invaluable in refining the quality of our work. Here are our responses to the points you raised:

  1. Dataset Selection in Section 3.1: We appreciate your inquiry about the choice of datasets in Section 3.1. Our decision to use KITTI exclusively for the occlusion handling section was intentional. The employed models in our study utilize either 2D, 3D, or both types of data for performance. NuScenes and OccludedPascal3D provide solely 3D data, and their datasets are not representative of real-world scenarios. To ensure a more comprehensive analysis of occlusion handling across various scenarios, we focused on KITTI, which provides both 2D and 3D data.
  2. Evaluation Metrics in Section 3.3: Your suggestion to include additional metrics such as mAP, Recall, and Precision is duly noted. We primarily used Average Precision (AP) as it encapsulates both Recall and Precision and is widely recognized as a comprehensive metric for object detection evaluation. Our emphasis on individual AP values for each object category allows for a more nuanced analysis, considering the distinct characteristics and challenges posed by different object types. In our future network proposal, we plan to incorporate additional metrics, including execution time, to offer a more comprehensive understanding of the experimental results.

We appreciate your diligence in reviewing our manuscript, and your feedback has significantly contributed to enhancing the clarity and robustness of our work.

Thank you once again for your time and valuable input.

Best regards,
Zainab Ouardirhi.

Reviewer 2 Report

Comments and Suggestions for Authors

The presented article is oriented towards intelligent video surveillance systems. In the introductory part, the authors processed a high-quality analysis of the available literature focused on the issue of intelligent systems. However, I miss there the analysis of the implemented tests, researches of other researchers, as it is an extensive topic.

Subsequently, the authors move on to the practical part, I miss a complex research methodology. In the practical part, the authors present several options for improving analytical functions in image processing. In this part, I am missing some more comprehensive own research, the authors' approach to the given issue. Of course, they list several options without any comprehensive comparison. I would have expected a better practical part in the article, and if the authors focus on several options, the entire comparison between the mentioned options should be made.

As part of the review, I would recommend adding such a comparison and at the same time adding a separate chapter - the conclusion, as I believe that both within the team plate of the journal there should be such a chapter and at the same time within it it should be stated how the authors see the possibilities of further research in the given issue.

Author Response

Greeting,

I hope this message finds you well. Thank you for taking the time to thoroughly review our manuscript. We appreciate your valuable feedback, which has undoubtedly contributed to enhancing the overall quality of our work.

  1. Introduction: We acknowledge your concern regarding the literature analysis in the introduction. To address this, we have expanded our review to include references that present test results of Smart Video Surveillance Systems (SVS) employing deep learning models. This addition aims to provide a more comprehensive overview of the current state of SVS technologies.
  2. Methodology and Practical Part: We understand your perspective on the need for a more intricate research methodology and a comprehensive comparison of the presented options. In response, we have refined the analysis methodology, particularly in the comparison section. We now provide a more detailed examination of the results, emphasizing the challenges faced by existing approaches during occlusion. Additionally, we have highlighted the limitations of each technique and emphasized the need for a tailored approach to occlusion handling.
  3. Conclusion: Your suggestion for a separate chapter in the conclusion aligns with our existing structure. We have a dedicated conclusion section that not only summarizes our findings but also outlines our perspectives and future directions for research in the given issue. We hope this addresses your concern and provides a clear roadmap for further exploration.

Once again, we sincerely appreciate your thoughtful insights, and we believe these revisions significantly strengthen the manuscript.

Thank you for your time and consideration.

Best regards,
Zainab Ouardirhi.

Reviewer 3 Report

Comments and Suggestions for Authors

This survey has discussed the progress of occlusion handling approches. There are some my concerns to it as below.

1. What is occlusion handling?

2. Need to explain existing survey work about the occlusion handling in introduction. Why your survey is different from existing work?

3. There is any Mathematical Model about the occlusion handling?

4. You can further explain concrete challenges in the occlusion handling algorithms? 

5. Future research directions should be drived under the concrete problems found by authers. Please authers clarify them.

Comments on the Quality of English Language

minor revision in the paper

Author Response

Greetings,

I hope this message finds you well, and I sincerely appreciate your thoughtful review of our manuscript. Your insights are invaluable, and I would like to address each of your concerns:

  1. Definition of Occlusion Handling: We acknowledge the importance of providing a concise explanation of occlusion handling. In the revised manuscript, we have included a brief definition to enhance the clarity of this crucial concept.
  2. Introduction and Distinction from Existing Surveys: Thank you for your suggestion to elaborate on existing survey work in the introduction. We have added further context to our introduction, emphasizing the distinct contributions of our survey. We’ve listed the enumerated contributions on the introduction, and they succinctly capture the unique aspects of our survey compared to existing literature.
  3. Mathematical Models for Occlusion Handling: Yes, indeed, there are mathematical models addressing occlusion handling, and we appreciate your pointing this out. Specifically, we cite the Probabilistic Occupancy Map (POM) approach, in the paper, contributing to the mathematical foundation of our study.
  4. Concrete Challenges in Occlusion Handling Algorithms: We have expanded our discussion on concrete challenges faced by occlusion handling algorithms. This addition aims to provide a more thorough understanding of the obstacles encountered in this domain.
  5. Future Research Directions and Identified Problems: In the revised manuscript, we have made an effort to further clarify our future research directions by explicitly connecting them to the concrete problems identified during our study. We hope this adds transparency and coherence to our proposed avenues for future research.

We genuinely appreciate your meticulous review, and your comments have significantly strengthened the overall quality of our manuscript.

Thank you for your time and valuable input.

Best regards,
Zainab Ouardirhi.

Round 2

Reviewer 1 Report

Comments and Suggestions for Authors

The answer of the authors to the dataset selection is confusing. It was said they focused on KITTI. However, in the experiments, the Pascal VOC 2012 dataset was used to explore alternative occlusion handling approaches. KITTI 2D and CityPersons datasets were used to evaluate the deep learning networks. Therefore, several datasets were used to different approaches. My suggestion is all occlusion handling approaches are evaluated by using the same datasets. Authors are recommended to provide enough data for complete comparison in Table 4.

Author Response

Thank you for your insightful feedback.

In light of your suggestion, if we grasp it correctly, we have revised Table 4 to include the alternative approaches referenced in Table 3. This modification allows for a unified evaluation of all occlusion handling methods using the same dataset, facilitating a comprehensive and direct comparison.

The updated table and corresponding analysis are now aligned with your suggestion, providing a clearer perspective on the performance of various approaches under consistent conditions. We believe this adjustment enhances the robustness and clarity of our comparative evaluation.

Please feel free to review the updated content in Table 4, and we welcome any further input you may have.

Cordially.